# Stability of Ion Exchange Membranes in Electrodialysis

**DOI:** 10.3390/membranes13010052

**Published:** 2022-12-31

**Authors:** Ksenia Solonchenko, Anna Kirichenko, Ksenia Kirichenko

**Affiliations:** 1Physical Chemistry Department, Faculty of Chemistry and High Technologies, Kuban State University, 149 Stavropolskaya St., 350040 Krasnodar, Russia; 2Department of Electric Engineering, Thermotechnics, Renewable Energy Sources, Faculty of Energetics, Kuban State Agrarian University named after I.T. Trubilin, 13 Kalinina St., 350004 Krasnodar, Russia

**Keywords:** ion exchange membrane, electrodialysis, fouling, scaling, matrix stretching, membrane discharge, Hofmann elimination, membrane operation, stability of membrane properties

## Abstract

During electrodialysis the ion exchange membranes are affected by such factors as passage of electric current, heating, tangential flow of solution and exposure to chemical agents. It can potentially cause the degradation of ion exchange groups and of polymeric backbone, worsening the performance of the process and necessitating the replacement of the membranes. This article aims to review how the composition and the structure of ion exchange membranes change during the electrodialysis or the studies imitating it.

## 1. Introduction

Since the 1950s, electrodialysis has been used industrially [1] for the purification, separation or concentration of substances. This method is based on the transport of ions through selectively permeable membranes under the action of an applied electric field [2]. Typically, ion exchange membranes, the materials that contain at least one ion exchange polymer bearing polar groups, are used in electrodialysis. In addition to the ion exchange polymer, ion exchange membranes may contain other components; additives that increase the mechanical strength of membranes [3] are most often used; however, other substances are also introduced, for example, nanotubes [4] or oxide nanoparticles that increase membrane selectivity [5,6]. The main types of ion exchange membranes are monopolar ones that have selectivity towards ions of a certain sign of charge, which are further subdivided into cation exchange membranes and anion exchange membranes, and bipolar ones, in which cation exchange and anion exchange components are simultaneously present and which are therefore impermeable to salt ions but are designed to generate H^+^ and OH^−^ ions at the boundary between the cation exchange and anion exchange components, which is called the bipolar boundary [7,8,9]. There are also asymmetric bipolar membranes that combine the function of generation of H^+^ and OH^−^ ions with the function of transporting salt ions [10,11,12,13,14]. Scheme of electrodialysis apparatus for desalination/concentration or for production of acid and alkali is depicted in Figure 1.

The cost of ion exchange membranes can be a significant part of the installation costs and, depending on the salinity of the treated water, a significant part of the total cost of the produced treated water [16]. In this regard, there is interest both in reducing the cost of membranes and in extending their service life. Unfortunately, during electrodialysis the membranes are affected by a number of factors that can lead to structural damage and degradation of membrane properties including the precipitation of organic and inorganic substances [17,18], shift of pH [19,20,21] and interaction with the components of the treated solution [22,23]. This review is devoted to the stability of ion exchange membranes during electrodialysis and considers the factors that act during electrodialysis and affect the properties of the membrane.

## 2. Fouling

### 2.1. Definition and Classification

Fouling [24] is the chemical interaction of the membrane with the components of the treated solution that leads to the deposition of some compounds, causing deterioration of the membrane properties. Usually, the deterioration of the properties of an ion exchange membrane is expressed as the loss of electrical conductivity [25] or in a decrease in the limiting current density; a model description of the decrease in the limiting current depending on the type of formed fouling film is presented in [26]. Fouling is recognized as one of the main problems of membrane technologies [27], and more publications are devoted to this problem than to other mechanisms of degradation of membrane properties during electrodialysis.

There are several variants for classifying the types of fouling depending on the causing agents, for example, Mikhaylin and Bazinet [28] distinguish (Figure 2) fouling with inorganic substances (which is denoted as scaling in some other articles), fouling with organic molecules and colloids (colloidal fouling of anion exchange membranes is discussed in a large number of publications, for example, in [29,30,31]; an example of the formation of peptide layers on a cation exchange membrane is described in [32]; peptide fouling of bipolar membranes is described in [33]) and biofouling, which, however, is more actively discussed in the context of reverse electrodialysis [34,35].

### 2.2. Scaling

Fouling with inorganic substances is also referred as scaling in some sources. Its characteristic manifestation is the deposition of inorganic substances on the surface or inside the pores of membranes with the formation of a “scaly” precipitate. When scaling occurs inside the pores of the membrane, the pores can be partially or completely blocked, which reduces the electrical conductivity; in addition, with the further course of the process, the pores can be stretched by sediment, which worsens the mechanical properties of the membrane. Precipitates are usually formed by CaCO_3_ [36], CaSO_4_ [26] and other insoluble sulfates (BaSO_4_, more rarely SrSO_4_ and RaSO_4_) [37], Ca(OH)_2_ and Mg(OH)_2_ [38], which are sparingly soluble compounds; therefore, it is noteworthy that that the formation of these precipitates can occur not only in the concentration chamber, but also in the desalination chamber. An example of the precipitates formed at the side of MA-41P membrane [39] in a solution of the following composition (Li^+^~0.25 g/L, Na^+^~1.15 g/L, K^+^~2.5 g/L, Ca^2+^~1.5 g/L, Mg^2+^~0.3 g/L, HCO_3_^−^~4.3 × 10^−3^ g/L, Cl^−^~8.86 g/L and pH 3.9) facing the desalination chamber is shown in Figure 3.

### 2.3. Organic and Colloid Fouling

Fouling is a particularly challenging issue in processing of solutions containing large organic molecules, examples of which are milk and whey, juices, wines [40] or natural waters which may contain impurities of humic acids that form chelate complexes with Ca^2+^ ions, irreversibly fouling the surface of membranes [41] (Figure 4) as well as of other natural organic matter capable of forming complexes with cations [42].

Fouling intensity depends on the compatibility of the chemical nature of membranes and contaminants: for example, since large molecules in the food industry are usually negatively charged, in treatment of solutions of food industry anion exchange membranes are more susceptible to fouling than cation exchange ones (as, for example, shown for the processing of food industry solutions containing polyphenols [43] and liquid digestate [44]); since large organic molecules in the food industry are mainly hydrophobic, hydrophobic membranes are more prone to fouling than hydrophilic ones [45]; if the cause of fouling is a compound with benzene rings, such as aromatic amino acid, phenol, anthocyanin or tannin, then polystyrene-divinylbenzene membranes are more susceptible to fouling due to the occurrence of π-stacking interactions between foulants and membranes [46,47] than the aliphatic membranes. In addition, even the result of colloidal fouling of paired cation exchange and anion exchange membranes in the same chamber of the electrodialyzer depends on the nature of the membrane: at the end of period of useful operation in the food industry both cation exchange and anion exchange membranes were degraded and both of them lost part of their exchange capacity, but cation exchange membranes became denser and their thickness decreased while the thickness of anion exchange membranes doubled [48].

In the case of colloidal fouling the formation of a bipolar boundary between the film and the membrane formed from the side of the desalination chamber also becomes a problem, which, aside from the typical consequences of fouling, causes an increase in the generation of H^+^ and OH^−^ ions and the issues related to this process, for example, an intensification of the precipitation of colloids of weak electrolytes [49]. Fouling of an anion exchange membrane and anion exchange resin with single-stranded DNA as an example of a large molecule carrying a negative charge was studied in [50], and it was shown that the fouling significantly changed the current–voltage curves of the membrane and that in the case of ion exchange resin the DNA changed the electrokinetic picture completely and very likely created bipolar junction capable of water splitting.

### 2.4. Fouling—Conclusions

An analysis of publications on the fouling of ion exchange membranes during electrodialysis shows that the topic of organic fouling, in particular colloidal fouling of anion exchange membranes, currently presents the greatest interest, which may be due to the increasing use of electrodialysis in the food industry and, consequently, due to chemistry of solutions in the food industry where large negatively charged molecules such as pectins. A much smaller number of publications is devoted separately to scaling, and a biofouling seems to fall within the scope of related processes such as reverse electrodialysis. At the same time, evidence appeared that a simultaneous presence of large organic molecules, especially natural organic matter, and polyvalent ions aggravates fouling, as it was shown for humic acid and Ca^2+^. This problem affects not only electrodialysis but also other membrane methods that deal with surface waters and groundwater and the solution of the problem is important for feasibility of salinity gradient power generation, so it can be expected that this topic would gain wider attention in future studies.

## 3. Other Mechanisms of Membrane Degradation

There are also mechanisms of degradation of ion exchange membranes that are not related to the deposition of substances on the surface or in the bulk of the membranes. The effect of physicochemical properties of treated solution such as high ionic strength and chemical reactions of components of the membrane with components of solution or with H^+^ and OH^−^ ions that may both be initially present in the solution and be generated during the electrodialysis in a reaction on the membrane surface catalyzed by its polar groups [8] cannot be reduced just to formation of sediments but has a significant effect.

### 3.1. Chemical Reactions

The most notable chemical reaction is concentration of H^+^ and OH^−^ ions in the treated solutions can be increased both due to the initially existing acidity or alkalinity of the treated solution. The problem is inherent for alkaline fuel cells that contain anion exchange membranes, where it received wide discussion [51], In addition, the generation of H^+^ and OH^−^ ions can be enhanced by the presence of ampholytes in the solution (in [52] such enhancement was studied for solutions of acid salts of weak acids).

The influence of H^+^ and OH^−^ ions is multifaceted. The pH shift caused by the presence of these ions leads to the loss of charge of weakly basic groups in strongly alkaline media (deprotonation of anion exchange membrane [53]) and by weakly acidic groups in strongly acidic media. The deprotonation of anion exchange membranes is much more often discussed than the discharge of cation exchange membranes, which, apparently, is associated with the greater prevalence of strongly dissociating sulfonic acid membranes compared to weakly acidic membranes, with the use of weakly basic membranes containing amino groups as well as the presence of amino groups even in strongly basic membranes [54]. The discharge of functional groups of monopolar membranes leads to loss of the membrane selectivity.

In addition to reversible reactions of protonation and deprotonation, H^+^ and OH^−^ ions are able to enter into irreversible reactions with the membrane material. The presence of OH^−^ ions is a much bigger issue. These ions are capable of reacting with ammonium bases and amino groups [55], as well as with sulfonium and phosphonium groups [56], which eliminate a carbon chain from the heteroatom (by mechanisms of Hofmann elimination [57], Stevens elimination and by other mechanisms [58], examples are given if Figure 5).

In terminal cases of these reactions the polar group is completely cleaved off [60], causing a loss of exchange capacity and an increase in membrane hydrophobicity, which in turn can enhance membrane fouling by hydrophobic molecules [61]. Such a reaction requires a high alkali concentration and heating (in the article [61], the degradation of groups containing quaternary nitrogen was studied in 3–9 M NaOH solution at 75 °C). In incomplete cases of transformation of nitrogen compounds quaternary ammonium bases are transformed into various amino groups [62]. Since amino groups are more catalytically active in the reaction of generation of H^+^ and OH^−^ ions [8], a positive feedback loop arises that accelerates the degradation of ammonium bases. Vasil’eva et al. studied the properties of a strongly basic membrane and showed that its quaternary ammonium bases are transformed into tertiary amino groups during electrodialysis [62].

Hydroxide ions can also damage the polymer matrix of the membrane. It was shown in [61,63] that OH^−^ ions reacts with polyvinyl chloride, which in the studied cases was represented by a reinforcing component of Neosepta AMX anion exchange membrane (Figure 6) and Neosepta CMX cation exchange membrane. The reaction splat off hydrogen chloride from the polymer and generated polyenes. These compounds lower the Young’s modulus of the membrane and increase its water content. In the aforementioned works it was shown that PVC of the Neosepta AMX membrane enters the cleavage reaction in much milder conditions compared to the PVC of the Neosepta CMX membrane, which is associated with the Donnan exclusion of hydroxide ions from the matrix of the cation exchange membrane.

It was reported in [64] that OH^−^ ions are able to interact with the layer-by-layer assembled membrane, the layers of which were anchored by hydrogen bonds. The interaction led to the destruction of the layered structure of the membrane, and it was proposed to stabilize the layer-by-layer membrane by sandwiching it between two electrospun membranes.

Membrane poisoning is close to fouling in an aspect that some substances get retained by the membrane; however, this term usually refers to the case when the substance does not form continuous layer or particles of sediments but form complexes with ion exchange groups or get absorbed by the membrane matrix instead. A typical example of these interaction is the blockage of ion exchange groups by polyvalent counterions [65,66]. 

### 3.2. Physical Interactions

In some cases, the components of the treated solution can damage the membrane structure even without chemical interactions.

One of the mechanisms for this is the stretching of the membrane matrix by ions with bulky hydration shells. Long-term electrodialysis of individual solutions containing strongly hydrated anions (NaH_2_PO_4_, and KC_4_H_5_O_6_ (potassium hydrotartrate)) and less hydrated inorganic anion (NaCl and NH_4_Cl) with Neosepta AMX and Neosepta AMX-Sb anion exchange membranes were studied in [67]. It was shown that after 300 h of electrodialysis the experimental limiting current increased by 33, 90 and 128% in the series NaCl < NaH_2_PO_4_ < KHT, respectively, and the authors attributed the growth to the destruction of C–C bonds of the ion exchange material, which occurs as a result of stretching the membrane matrix by strongly hydrated ions. The work [68] also reports the degradation of anion exchange membranes with the loss of their functional groups and the formation of cavities on the membrane surface upon treatment of a solution containing phosphate and sulfate ions (Figure 7).

The second reason for degradation without chemical interactions is an increase in the ionic strength of the solution, which in turn suppresses the electrostatic interactions between the components of the membrane, for example, between the substrate membrane and selective layers. In [69], the devolution of the properties of a monovalent selective membrane during electrodialysis of brines from Llullaillaco Salt Lake (Argentina), in which the total concentration of dissolved solids (409 g/L TDS) was very high, was studied. The investigation demonstrated the degradation of the surface nanostructure and charge functionalization, caused, according to the authors, by a decrease in the electrostatic interaction between the polyethylenimine active layer and the polystyrene–divinylbenzene substrate due to high ionic strength of the treated solution and resulting in a decrease in monovalent selectivity from 16.6 to 5.67 was shown.

### 3.3. Conclusion to Mechanisms of Degradation

Ion exchange membrane stability may be compromised not only by reversible and partially reversible mechanisms such as fouling but also by other mechanisms that damage the integrity of the membrane and hence are irreversible. This problem is more frequently considered in relation to membrane electrolysis, fuel cells and redox flow batteries where harsher conditions such as presence of active oxidizing species, heating and low humidity are present, but in some conditions these problems are encountered in electrodialysis as well, for example in bipolar membrane electrodialysis and at overlimiting current modes of monopolar membrane electrodialysis, where the membrane itself becomes the source of OH^−^ ions.

The problem of damage to membrane structure has further importance since it follows from the reviewed publications that the processes of membrane degradation are interrelated and can be mutually enhanced. The example of such enhancement is shown in Figure 8.

Hence effective prevention of one type of membrane degradation requires the control over other mechanisms of membrane degradation as well.

## 4. Approaches to Prevention of Degradation

The methods of prevention or reversal of membrane degradation are even more numerous than the mechanisms of membrane fouling, and the new techniques are constantly proposed. Enhancement of mechanisms of degradation of ion exchange membranes by the action of other mechanisms calls for techniques that can prevent or reverse several mechanisms. Several possible approaches are discussed in short below. 

### 4.1. Pretreatment of Solution

If simultaneous presence of several fouling agents presents the biggest problem, then the pretreatment of solutions that would remove one of these components might be the key. For example, calcium might be removed from the solutions using pellet reactor pretreatment [70].

Pretreatment is usually carried out through dosing of chemicals [71], through adsorption on some substrate [72,73], through centrifugation or introduction of filtration steps [74,75,76,77]. Addition of steps to treatment procedure adds costs. Dosing of chemicals limits the ecological advantages of electrodialysis technique. In food industry filtration pretreatment may remove valuable compounds from the treated products, which, however, might be reintroduced at later stages [75].

Despite the disadvantages discussed above, in some cases the pretreatment is currently absolutely necessary, for example, in wastewater valorization by electrodialysis where high organic matter load causes high probability of fouling [72] (Figure 9) or other processes with high organic load [74].

### 4.2. Optimization of Electric Current Modes

The first industrial application of electrodialysis dates back to 1954, but the membranes were initially subject to severe fouling, which required the process to be stopped to flush the membranes, which led to the formation of secondary wastewater. The situation changed with the development of electrodialysis reversal [1,77], in which the polarity of the electrodes was periodically changed, and the hydraulic regime was simultaneously changed, as a result of which the former desalination chambers became concentration chambers, and vice versa. During electrodialysis reversal, some foulants are removed from the membrane surface and pores; however, some contaminants may be too strongly bound to the membrane surface or form stable fouling layers that will not “purge” from the membrane after polarity reversal process [78]. This prompted the search for new ways to prevent and to reverse fouling.

Not only polarity reversal can reduce the intensity of fouling, but also the use of pulsed electric fields. It was shown in [79,80] that the use of a pulsed electric field can significantly reduce precipitation (Figure 10) due to the reduced concentration polarization due to relaxation of concentration profiles during the pause interval.

### 4.3. Tailoring Hydrodynamic Conditions

One of the most powerful tools for prevention of multiple mechanisms of membrane degradation is tailoring of the hydrodynamic conditions [81]. The solution flow rate and hydrodynamics in general determine concentration polarization, and with it such significant processes as generation of H^+^ and OH^−^ ions and fouling. In turn, the hydrodynamics is determined by the shape of the channel and by the presence of spacer [82]. As expected, the enhancement of turbulence within the chambers of electrodialysis apparatuses helps washing out the deposits [83,84] and also removing the H^+^ and OH^−^ ions produced in catalytic generation reaction. Spacers of advanced geometry [85] or novel composition (e.g., ion-conductive spacers [86]) are used to promote the development of turbulence. 

However, nonconductive spacers have a disadvantage of so-called shadow effect on the membrane and on the solution—the covering of the active area of the membrane and making ions transport in tortuous paths, respectively [87]. Covering of the active area can be avoided by the transition from the flat membranes separated by a spacer to a profiled membranes in an empty channel [88] (the intermembrane distance is maintained by the alignment of the profile elements [89], which is one of the challenges of use of profiled membranes). For reverse electrodialysis, it is shown that the transition from spacers to profiled membranes reduces fouling [90] (Figure 11).

### 4.4. Surface Modification of Membrane

One of the methods of suppression of undesirable phenomena in membrane system is the modification of the membrane surface.

In [91], to prevent fouling of anion-exchange membranes by anionic surfactants, carboxyl groups were created on the membrane surface by oxidation treatment. In the dissociated state the carboxyl groups possess a charge, the sign of which is the same as that of anionic surfactants and opposite to that of the polar groups in the membrane bulk. Thus, an increase in the resistance to fouling of the membrane modified with polar groups, the sign of charge of which is opposite to the sign of charge of the polar groups in the membrane bulk, is shown.

The deposition of a bifunctional polymer containing quaternary ammonium bases on the surface of an anion exchange membrane to reduce the generation of H^+^ and OH^−^ ions was shown in [92]. Electrochemical impedance spectroscopy showed that the modification is stable for 50 h, after which the separation of the modifier begins, which in turn enhances the generation of H^+^ and OH^−^ ions. Thus, an increase in the resistance to fouling of the membrane modified with polar groups sign of charge of which is the same as the sign of charge of the polar groups within the membrane bulk is shown.

Finally, in [93], an increase in the resistance of a membrane layer-by-layer-coated with polymers bearing polar groups with alternating signs of charges (in this case, it was polyallylamine and sodium polystyrene sulfonate) to fouling by a sodium dodecylbenzene sulfonate was shown (Figure 12). As an explanation, increased hydrophilicity and electrostatic interactions with the negatively charged membrane surface (in the case of membranes whose top layer was sodium polystyrene sulfonate) have been proposed.

In addition, the intensity of fouling increases with increasing surface roughness [94], which suggests that membrane modifications aimed at reduction of surface roughness will also suppress membrane fouling (apparently this mechanism played a role in increase of the scaling resistance of heterogeneous membranes after deposition of homogenizing layer [38]).

### 4.5. Crosslinking

It has been shown that the modification with the introduction of a hydrophobic crosslinking agent into the anion exchange membrane makes it possible to simultaneously achieve high electrical conductivity and a low swelling ratio [95]. On the other hand, for other processes, an increase in the resistance to fouling after the introduction of a hydrophilic crosslink was shown in [96], where it was also found that the membrane with the highest fraction of the introduced crosslinking agent possesses the highest resistance to fouling. Hence, it might be concluded that increased crosslinking improves the fouling resistance regardless of composition of crosslinking agent. However, crosslinking shifts the balance of the selectivity–permeability tradeoff to the selectivity side [97], and the conductivity of such membranes decreases.

### 4.6. Membrane Regeneration

Regeneration procedures is a way to restore fouled or poisoned membrane to their pristine state. The chemicals used for regeneration tend to depend on the nature of the fouling/poisoning agent. It is understandable that the scaling by CaCO_3_ can be reduced by HCl treatment [98] and the membranes that treated cyanide-free brass electrodeposition effluents that contain zinc and copper are regenerated with NaOH solutions [99] (it is also noteworthy that the article reports that increasing concentration of alkali by itself causes degradation of anion exchange membranes). Article [98] also reports that NaOH was more efficient at removing oils from both anion exchange membrane and from cation exchange membrane.

The organic and colloid fouling is usually cleaned by concentrated solution of salts (to disrupt the electrostatic interactions between the molecules of foulant and facilitate their removal) or by alcohols or their mixture with water. The article [100] explored methods for cleaning cation exchange membranes used in the food industry that included soaking into one of the following solutions: NaCl with a concentration of 35 g/L, reconstituted seawater or a water–ethanol mixture, and showed that the best results are demonstrated by cleaning with the water–ethanol mixture, but with this treatment the fraction of electroneutral solution within the membrane increases with time.

Additionally, the membranes operating in food industry treating the solutions with high content of nutrients may require antimicrobial treatment in addition to the fouling control. It can be achieved by cleaning with sodium hypochlorite, but the prolonged contact with it also causes the degradation of the membrane [101].

### 4.7. Conclusions to Prevention of Membrane Degradation and Membrane Cleaning

Prevention and control of membrane degradation is one of the most studied topics in electrodialysis and other membrane technologies with a multitude of suggested mechanisms. The perfect solution that would allow completely negating fouling (still) does not exist, and it should be noted that several methods aimed at prevention of degradation cause the other form of membrane degradation themselves, which is especially noticeable in the case of alkali cleaning of anion exchange membranes. Damage to the membrane properties during cleaning and the demand for greener production with lower number of added chemicals cause interest in reagentless approaches to prevention of degradation, since the greater attention to electric field modes and hydrodynamics of membrane channel, as well as to novel approaches to membrane modification that would provide a long-lasting result without the need in repeated treatments.

## 5. Conclusions and Perspectives

The stability of ion exchange membranes presented a major challenge for electrodialysis since the introduction of this process, and to date, the issues of fouling of the deterioration of the polymer structure of membranes and of the loss of ion exchange groups are an important direction for research and development efforts.

With all the variety of mechanisms of arising processes and their external manifestations that include an increase or decrease in the thickness of the membrane, an increase in the hydrophilicity or hydrophobicity of membranes, the formation of films on the surface of membranes or crystals in their bulk, the result of these processes is very similar. At the end of the service life, the structure of the membrane is disturbed to some extent, the exchange capacity, electrical conductivity and the limiting current of salt ions through the membrane are reduced. The mechanisms for prevention of fouling are very similar and one action can prevent several mechanisms of degradation from occurrence. The same approaches to the prevention of several types of membrane degradation have the advantage of possibility to deal with several mechanisms of membrane devolution simultaneously.

An analysis of publications on the stability of ion exchange membranes in electrodialysis also shows that the majority of the publications are focused on specific water treatment processes that usually operate with commercial membranes (Neosepta AMX and Neosepta CMX manufactured by Astom, MK-40 and MA-41 produced by Shchekinoazot), while it seems promising to study the stability of new membranes, for which improved properties are assumed, in order to assess the retention of improved properties during membrane operation.

## Figures and Tables

**Figure 1 membranes-13-00052-f001:**
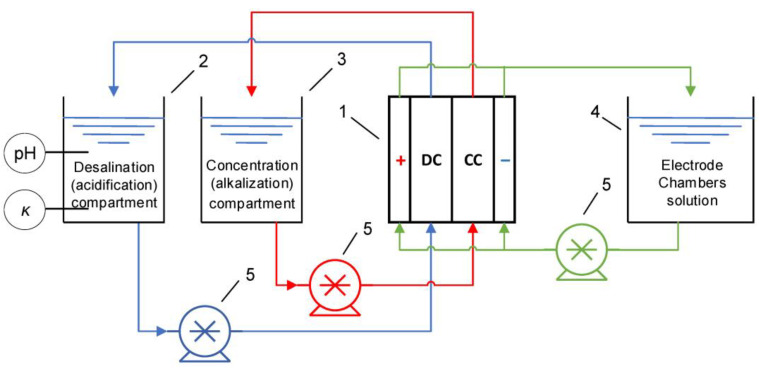
Hydraulic scheme of the electrodialysis setup: 1—electrodialysis module, 2—desalination (acidification) compartment tank, 3—concentration (alkalization) compartment tank, 4—electrode rinse solution tank, 5—pumps. DC denotes desalination chamber and CC denotes concentration chamber. Reprinted with permission from Ref. [15] under the terms of the Creative Commons Attribution 4.0 International License (http://creativecommons.org/licenses/by/4.0/ (accessed on 26 December 2022)). Copyright © 2022 by the authors. Licensee MDPI, Basel, Switzerland.

**Figure 2 membranes-13-00052-f002:**
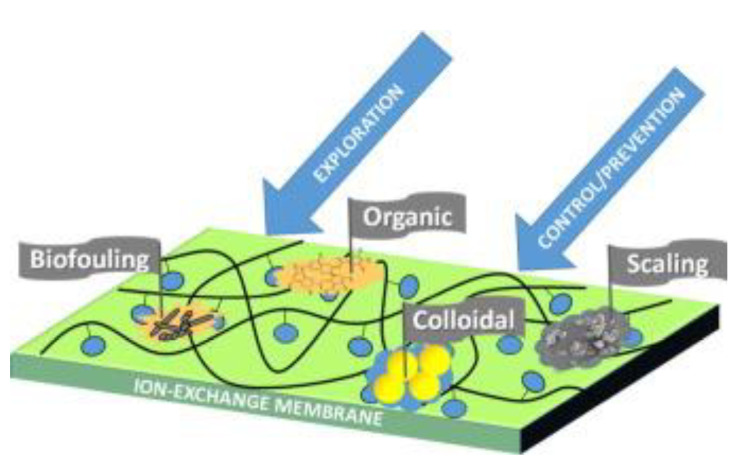
Types of fouling. Reprinted with permission from [28]. Copyright © 2015 Elsevier B.V.

**Figure 3 membranes-13-00052-f003:**
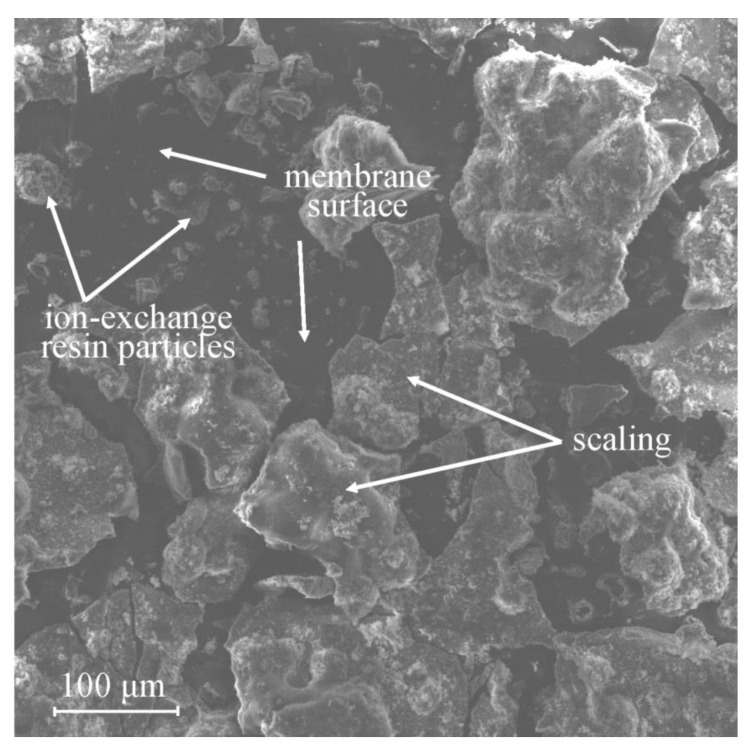
Scaling of the MA-41P membrane in a solution containing calcium and magnesium ions. Reprinted with permission from Ref. [39] under the terms of the Creative Commons Attribution 4.0 International License (http://creativecommons.org/licenses/by/4.0/ (accessed on 26 December 2022)). Copyright © 2022 by the authors. Licensee MDPI, Basel, Switzerland.

**Figure 4 membranes-13-00052-f004:**
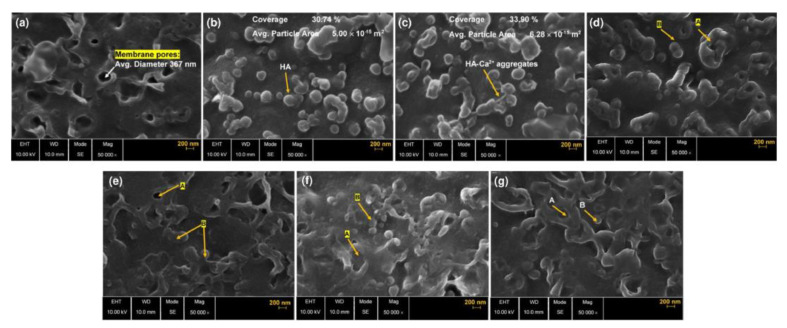
High-resolution scanning electron microscopy images illustrating surface morphology of (**a**) activated AEM (anion exchange membrane) after immersing the virgin membrane in 0.001 M NaCl for 24 h, (**b**) AEM fouled by HA (humic acid) and (**c**) AEM fouled by HA-Ca^2+^. Aggregated HA-Ca2+ foulants on/in AEM pores are labeled, (**d**) 2% (*w*/*w*) NaCl-desorbed, HA-fouled membrane and (**e**) 2% (*w*/*w*) NaCl-desorbed, HA-Ca2+-fouled membrane, (**f**) 0.03% (*w*/*w*) SDS (sodium dodecyl sulfonate)-desorbed, HA-fouled membrane and (**g**) 0.03% (*w*/*w*) SDS-desorbed, HA-Ca^2+^-fouled membrane. Label A presents recovered AEM pores by desorption and label B shows the remaining foulant after desorption. Reprinted with permission from Ref. [41] under the terms of the Creative Commons Attribution 4.0 International License (http://creativecommons.org/licenses/by/4.0/ (accessed on 26 December 2022)). Copyright © 2021 Published by Elsevier B.V.

**Figure 5 membranes-13-00052-f005:**
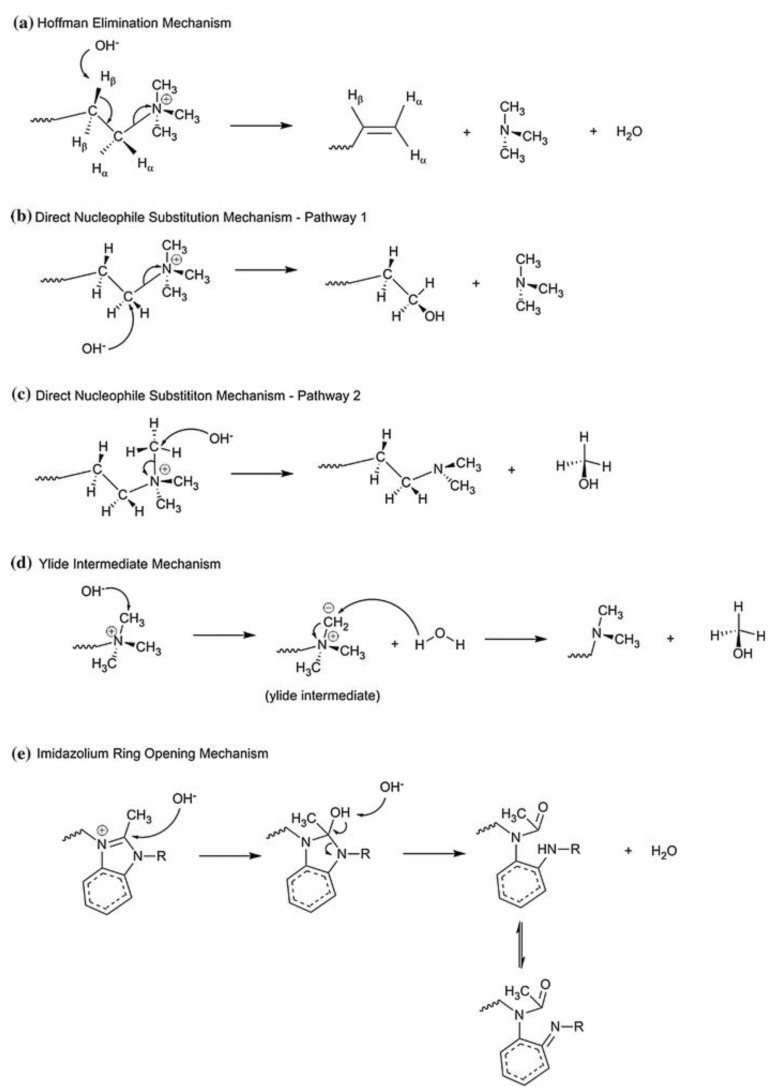
Scheme of the mechanisms of degradation of ammonium bases. Reproduced with permission from Ref. [59] under the terms of the Creative Commons Attribution 4.0 International License (http://creativecommons.org/licenses/by/4.0/ (accessed on 26 December 2022)). Copyright © 2018, the author(s).

**Figure 6 membranes-13-00052-f006:**
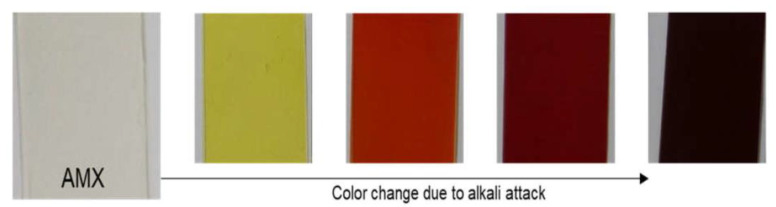
A color change trend of commercial AMX during the alkali immersion test. Reprinted with permission from Ref. [63] under the terms of the Creative Commons Attribution 4.0 International License (http://creativecommons.org/licenses/by/4.0/ (accessed on 26 December 2022)). Copyright © 2018 by the authors. Licensee MDPI, Basel, Switzerland.

**Figure 7 membranes-13-00052-f007:**
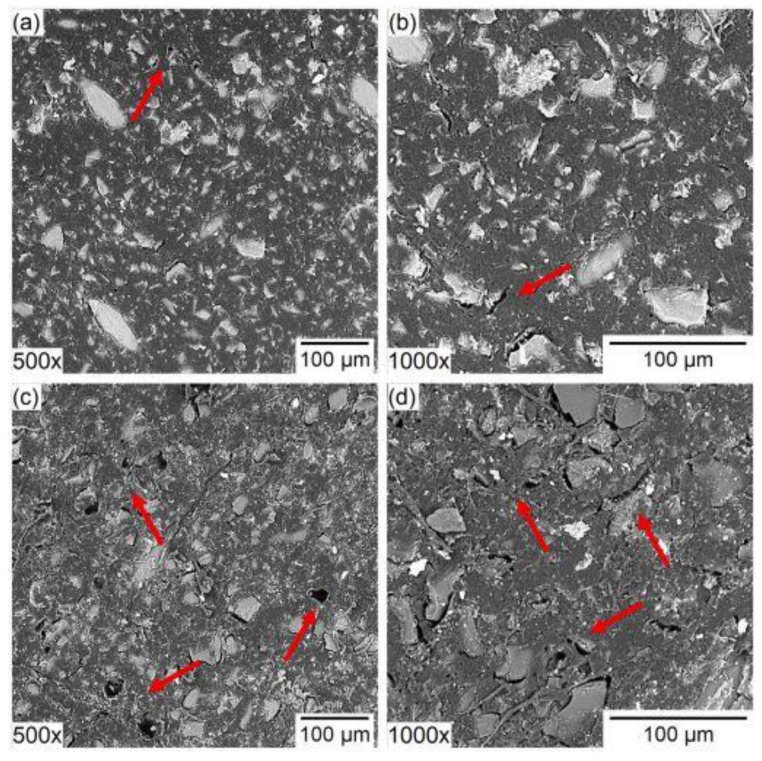
Scanning electron microscopy images of the surface of (**a**,**b**) original and (**c**,**d**) used anion-exchange membrane 500× and 1000× of magnification. Reprinted with permission from Ref. [68]. Copyright © 2021 Elsevier B.V.

**Figure 8 membranes-13-00052-f008:**
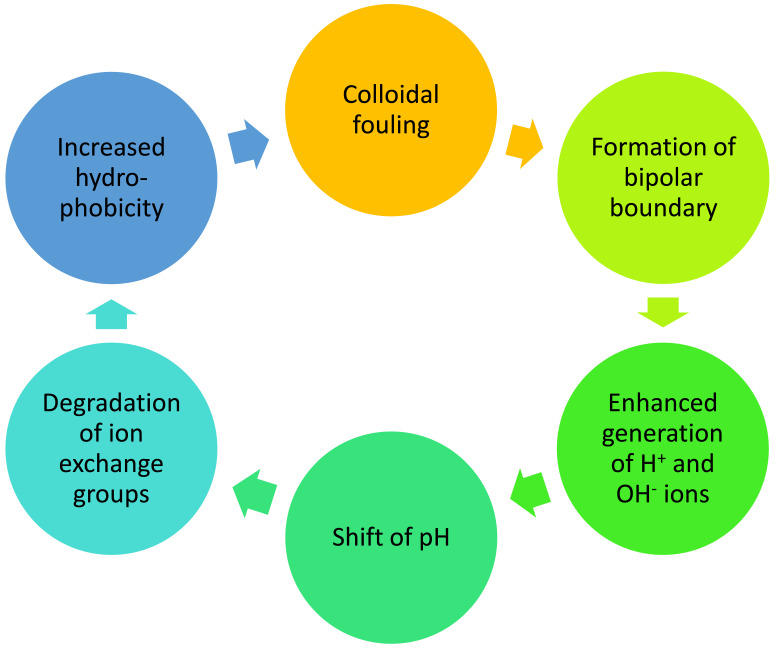
Example of processes of membrane degradation enhancing each other in a cycle.

**Figure 9 membranes-13-00052-f009:**
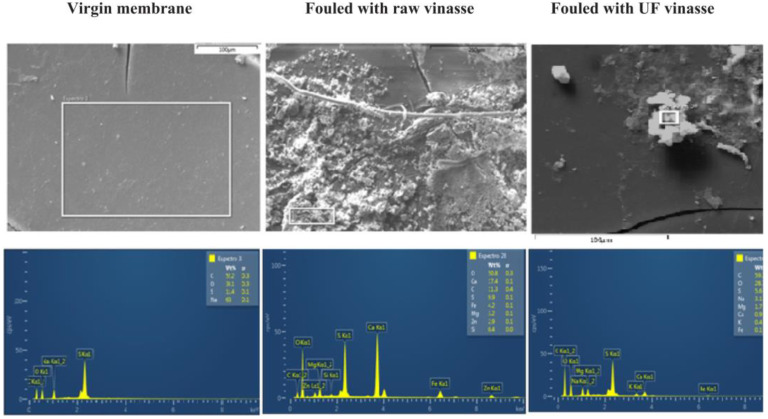
SEM micrographs and EDS (energy dispersive spectroscopy) analysis of cationic virgin membrane, membrane fouled with raw vinasse, and membrane fouled with ultrafiltrated vinasse. Reprinted with permission from Ref. [74]. Copyright © 2019 Elsevier B.V.

**Figure 10 membranes-13-00052-f010:**
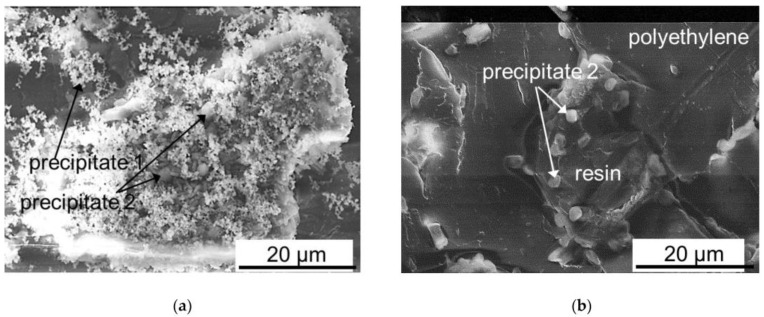
Scanning electron microscopy images of mineral precipitates on granules of anion-exchange resin protruding on the surface of MA-41P after ED demineralization of a multicomponent solution in CC2 (continuous current) (**a**) and PEF (pulsed electric field) (**b**) modes. Reprinted from Ref. [80] under the terms of the Creative Commons Attribution 4.0 International License (http://creativecommons.org/licenses/by/4.0/ (accessed on 26 December 2022)). Copyright © 2022 by the authors. Licensee MDPI, Basel, Switzerland.

**Figure 11 membranes-13-00052-f011:**
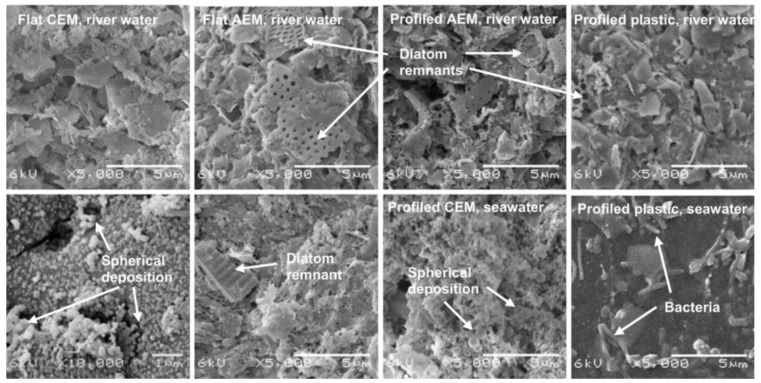
SEM images of cation and anion exchange membranes and profiled plastics. The images in the upper row were in contact with river water, whereas the images in the lower row where in contact with seawater. The four images at the left side are obtained from flat membranes (stack 1, river water side and seawater side), the four images at the right side are obtained from profiled membranes (stack 2) and profiled plastics (stack 3). Most images were saved at a magnification of 5000×, only the flat (cation exchange membrane) in contact with seawater was saved at 10,000×, to see the spherical deposition. Remnants of diatoms, bacteria and spherical deposition are indicated by arrows. Reprinted with permission from Ref. [90]. Copyright © 2012 Elsevier Ltd.

**Figure 12 membranes-13-00052-f012:**
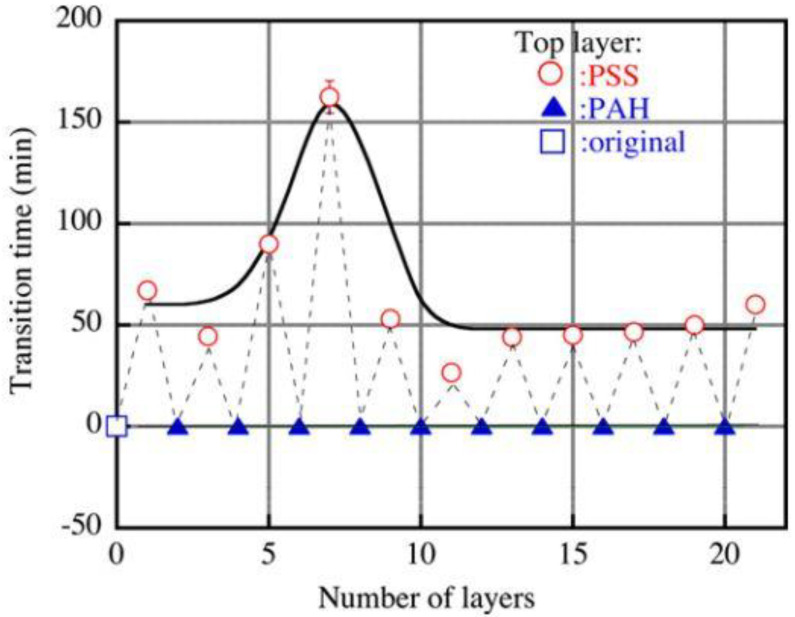
Effect of the number of layers on the transition time (in this case, the time elapsed before the onset of fouling). The feed solution contained 0.05 M NaCl and 0.052 kg/m^3^ SDBS. Squares represent the original membrane, circles represent the membrane terminated with PSS (polystyrene sulfonate), and triangles represent the membrane terminated with PAH (polyallylamine). The modification solution contained 0.3 kg/m^3^ of PSS or PAH, and 1 M NaCl. Reprinted with permission from Ref. [93]. Copyright © 2012 Elsevier B.V.

## Data Availability

The article does not contain new data.

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
