# Peer review of "Stability of Ion Exchange Membranes in Electrodialysis"

_membranes, 2022, doi:10.3390/membranes13010052_

Round 1
Reviewer 1 Report
This manuscript gives some information about membrane degradation during electrodialysis. However, it requires serious modification.
1. This review article contains many references to other review articles.
2. The material should be summarized using tables and charts.
3. A rather superficial analysis of materials related to the degradation of ion-exchange membranes is given. Such an analysis should be carried out in more depth.
4. This review contains only two figures, and it is rather strange for Review articles. Figure 1: the need for its use is not clear, because the article does not consider different options for hydraulic circuits of electrodialysis plants. The Figure 2 shows the deposition of inorganic ions on the membrane surface. For sure there are publications where one can find the deposition of organic substances on the surface, as well as photographs depicting changes in the surface or volume of membranes under the influence of other factors.
5. Despite the fact that there are the sections in the article, sometimes it is hard to figure out. Probably a different structuring would have helped here. Or an illustration of the text with the involvement of generalizing schemes and figures.
6. Line 35. "Scheme of variants of electrodialysis apparatuses is depicted in Figure 1." However, Fig. 1 shows the Hydraulic scheme of the electrodialysis setup without detailing the options for placing different types of membranes in electrodialysis machines. The title of the Figure does not match what is shown on it. The numbering of the components shown in the diagram is missing.
7. Lines 43-47 It is hard to believe that only works 17-19 describe these common degradation variants, there are probably many more such works. In addition, 17-19 are review articles. What is the specificity of the proposed review?
8. Lines 86-89 are out of line. Above in the text, the stability time was not given anywhere.
9. Contents of the section “5.5. Pretreatment and regeneration" is not disclosed. What preprocessing methods are there? What is the peculiarity of the regeneration of membranes used in the processing of solutions of organic substances, in contrast to inorganic, etc. There is no connection between this section and the types of membrane degradation identified by the authors.
10. When describing the effect of membrane pretreatment, there is a reference to an article [78] in which the Aspire® QL217 membrane is used in the process of membrane distillation. Is this membrane ion exchange? Will the behavior of membranes be similar during electrodialysis? After all, this review is devoted to studying the stability of ion-exchange membranes during electrodialysis. The same questions may also arise referencing the article [79], where the process of nanofiltration is considered. Why the review about ion-exchange membranes provide data on the effectiveness of pre-treatment of nanofiltration membranes? Are they made of the same material, or are they ion exchangers? There are also a number of references that are devoted to other membrane methods, and not to processes with ion-exchange membranes: [77, 76, 74].
In the introduction, it is said that nanotubes can be introduced into ion-exchange membranes; the authors give a reference [4], where sensors based on such a membrane are considered. Meanwhile, the use of ion-exchange membranes modified with nanotubes can also be found for electrodialysis. This will more clearly correspond to the topic of the article.
Author Response
Reviewer 1.
This manuscript gives some information about membrane degradation during electrodialysis. However, it requires serious modification.
- This review article contains many references to other review articles.
Response: part of references was replaced in response to other comments, and we tried to focus on research articles with replacements.
- The material should be summarized using tables and charts.
Response: new figures and charts were added.
- A rather superficial analysis of materials related to the degradation of ion-exchange membranes is given. Such an analysis should be carried out in more depth.
Response: we thank the reviewer for the suggestion, the changes made for article were done with it in mind. Additionally now we have the conclusions for major sections.
- This review contains only two figures, and it is rather strange for Review articles. Figure 1: the need for its use is not clear, because the article does not consider different options for hydraulic circuits of electrodialysis plants. The Figure 2 shows the deposition of inorganic ions on the membrane surface. For sure there are publications where one can find the deposition of organic substances on the surface, as well as photographs depicting changes in the surface or volume of membranes under the influence of other factors.
Response: a number of figures was added, including the deposition of organic substances on the membrane surface and a photographs depicting changes under the influence of other changes.
- Despite the fact that there are the sections in the article, sometimes it is hard to figure out. Probably a different structuring would have helped here. Or an illustration of the text with the involvement of generalizing schemes and figures.
Response: we reorganized the subdivision on sections and added a generalizing scheme
- Line 35. "Scheme of variants of electrodialysis apparatuses is depicted in Figure 1." However, Fig. 1 shows the Hydraulic scheme of the electrodialysis setup without detailing the options for placing different types of membranes in electrodialysis machines. The title of the Figure does not match what is shown on it. The numbering of the components shown in the diagram is missing.
Response: restored the original numbering (and added the comment on the meaning of DC and CC).
- Lines 43-47 It is hard to believe that only works 17-19 describe these common degradation variants, there are probably many more such works. In addition, 17-19 are review articles. What is the specificity of the proposed review?
Response: We surely agree that much more works were done in this area, we added more references here and replaced references 17 and 19. Additionally more references were given below in the corresponding subsections.
- Lines 86-89 are out of line. Above in the text, the stability time was not given anywhere.
Response: Agreed on the stability time, reworded to make it more consistent. Unfortunately I have to inquire on the meaning of “out of line”, since if it is about formatting, then it has “MDPI figure caption” style which differs from the main text.
- Contents of the section “5.5. Pretreatment and regeneration" is not disclosed. What preprocessing methods are there? What is the peculiarity of the regeneration of membranes used in the processing of solutions of organic substances, in contrast to inorganic, etc. There is no connection between this section and the types of membrane degradation identified by the authors.
Section “pretreatment and regeneration” was split and two and rewritten. We thank the reviewer for the prompts on the content of the new sections.
- When describing the effect of membrane pretreatment, there is a reference to an article [78] in which the Aspire® QL217 membrane is used in the process of membrane distillation. Is this membrane ion exchange? Will the behavior of membranes be similar during electrodialysis? After all, this review is devoted to studying the stability of ion-exchange membranes during electrodialysis. The same questions may also arise referencing the article [79], where the process of nanofiltration is considered. Why the review about ion-exchange membranes provide data on the effectiveness of pre-treatment of nanofiltration membranes? Are they made of the same material, or are they ion exchangers? There are also a number of references that are devoted to other membrane methods, and not to processes with ion-exchange membranes: [77, 76, 74].
Reference 74 was replaced. The references from the former “pretreatment and regeneration” section were completely replaced since the section was rewritten.
In the introduction, it is said that nanotubes can be introduced into ion-exchange membranes; the authors give a reference [4], where sensors based on such a membrane are considered. Meanwhile, the use of ion-exchange membranes modified with nanotubes can also be found for electrodialysis. This will more clearly correspond to the topic of the article.
Response: we found several references of modification of ion exchange membranes intended for electrodialysis with nanotubes and replaced the reference with [Fan, H.; Huang, Y. and Yip, N.Y. Advancing the conductivity-permselectivity tradeoff of electrodialysis ion-exchange membranes with sulfonated CNT nanocomposites. J. Memb. Sci. 2020, 610, 118259. DOI: 10.1016/j.memsci.2020.118259]
Reviewer 2 Report
1. What is the meaning of CC, DC and the number from 1 to 5 in Figure 1.
2. When the abbreviation first appears, it needs to be given its full name. For example, PEI and PS-DVB in line 201.
3. The “+” in H+ and “-“ in OH- should be superscripted.
4. More information should be given in section of 5.2.
5. A conclusion should be drawn at the end of every section. The review is not just a simple introduction of what each previous article is doing, but to draw an effective conclusion from the previous research.
6. The section of 5.5 should be rewritten.
7. The conclusion is too tedious.
Author Response
- What is the meaning of CC, DC and the number from 1 to 5 in Figure 1.
Response: the caption now reads as “Hydraulic scheme of the electrodialysis setup. 1—electrodialysis module, 2—desalination (acidification) compartment tank, 3—concentration (alkalization) compartment tank, 4—electrode rinse solution tank, 5—pumps. DC denotes desalination chamber and CC denotes concentration chamber. Reproduced from [Nosova et al.].”
- When the abbreviation first appears, it needs to be given its full name. For example, PEI and PS-DVB in line 201.
Response: the abbreviations were dealt with (those two were scrapped and new ones inevitably appearing due to their presence in added figures were given the full name in brackets)
- The “+” in H+ and “-“ in OH- should be superscripted.
Response: found and fixed a couple of non-superscripted ones.
- More information should be given in section of 5.2.
Response: the former section 5.2 is expanded as follows:
Tailoring hydrodynamic conditions
One of the most powerful tools for prevention of multiple mechanisms of membrane degradation is tailoring of the hydrodynamic conditions [76]. The solution flow rate and hydrodynamics in general determine concentration polarization, and with it such significant processes as generation of H+ and OH- ions and fouling. In turn, the hydrodynamics is determined by the shape of the channel and by the presence of spacer [77]. As expected, the enhancement of turbulence within the chambers of electrodialysis apparatuses helps washing out the deposits [78,79] and also removing the H+ and OH- ions produced in catalytic generation reaction. Spacers of advanced geometry [80] or novel composition (e.g. ion conductive spacers [81]) are used to promote the development of turbulence.
However, nonconductive spacers have a disadvantage of so called shadow effect on the membrane and on the solution – the covering of the active area of the membrane and making ions transport in tortuous paths, respectively [82]. Covering of the active area can be avoided by the transition from the flat membranes separated by a spacer to a profiled membranes in an empty channel [83] (the intermembrane distance is maintained by the alignment of the profile elements [84], which is one of the challenges of use of profiled membranes). For reverse electrodialysis it is shown that the transition from spacers to profiled membranes reduces fouling [85] (Figure 11).
Figure 11. SEM images of cation and anion exchange membranes and profiled plastics. The images in the upper row were in contact with river water, whereas the images in the lower row where in contact with seawater. The four images at the left side are obtained from flat membranes (stack 1, river water side and seawater side), the four images at the right side are obtained from profiled membranes (stack 2) and profiled plastics (stack 3). Most images were saved at a magnification of 5000×, only the flat [cation exchange membrane] in contact with seawater was saved at 10,000×, to see the spherical deposition. Remnants of diatoms, bacteria and spherical deposition are indicated by arrows. Reproduced from [85] with permission.
- A conclusion should be drawn at the end of every section. The review is not just a simple introduction of what each previous article is doing, but to draw an effective conclusion from the previous research.
Response: a conclusion is now drawn at the end of every major section. It additionally solved the problem of too tedious final conclusions.
- The section of 5.5 should be rewritten.
Response: the section 5.5 was split and rewritten.
- The conclusion is too tedious.
Response: the conclusion was trimmed
Round 2
Reviewer 1 Report
The authors have improved the manuscript.